# Effects of Softening Dry Food with Water on Stress Response, Intestinal Microbiome, and Metabolic Profile in Beagle Dogs

**DOI:** 10.3390/metabo12111124

**Published:** 2022-11-16

**Authors:** Limeng Zhang, Kang Yang, Shiyan Jian, Zhongquan Xin, Chaoyu Wen, Lingna Zhang, Jian Huang, Baichuan Deng, Jinping Deng

**Affiliations:** 1Maoming Branch, Guangdong Laboratory for Lingnan Modern Agriculture, Guangdong Provincial Key Laboratory of Animal Nutrition Control, National Engineering Research Center for Breeding Swine Industry, College of Animal Science, South China Agricultural University, Guangzhou 510642, China; 2Research Center of Pet Nutrition, Guangzhou Qingke Biotechnology Co., Ltd., Guangzhou 510642, China; 3Institute for Quality & Safety and Standards of Agricultural Products Research, Jiangxi Academy of Agricultural Sciences, Nanchang 330200, China

**Keywords:** water-softened dry food, pet food, beagle dog, inflammatory response, fecal microbiota, metabolomics

## Abstract

Softening dry food with water is believed to be more beneficial to the intestinal health and nutrients absorption of dogs by some owners, but there appears to be little scientific basis for this belief. Thus, this study aimed to compare feeding dry food (DF) and water-softened dry food (SDF) on stress response, intestinal microbiome, and metabolic profile in dogs. Twenty healthy 5-month-old beagle dogs were selected and divided into two groups according to their gender and body weight using a completely randomized block design. Both groups were fed the same basal diet, with one group fed DF and the other fed SDF. The trial lasted for 21 days. The apparent total tract digestibility (ATTD) of nutrients, inflammatory cytokines, stress hormones, heat shock protein-70 (HSP-70), fecal microbiota, short-chain fatty acids (SCFAs), branch-chain fatty acids (BCFAs), and metabolomics were measured. Results showed that there was no significant difference in body weight, ATTD, and SCFAs between the DF and SDF groups (*p* > 0.05), whereas feeding with SDF caused a significant increase in serum cortisol level (*p* < 0.05) and tended to have higher interleukin-2 (*p* = 0.062) and HSP-70 (*p* = 0.097) levels. Fecal 16S rRNA gene sequencing found that the SDF group had higher alpha diversity indices (*p* < 0.05). Furthermore, the SDF group had higher levels of *Streptococcus*, *Enterococcus*, and *Escherichia_Shigella*, and lower levels of *Faecalibacterium* (*p* < 0.05). Serum and fecal metabolomics further showed that feeding with SDF significantly influenced the purine metabolism, riboflavin metabolism, and arginine and proline metabolism (*p* < 0.05). Overall, feeding with SDF caused higher cortisol level and generated effects of higher intestinal microbial diversity in dogs, but it caused an increase in some pathogenic bacteria, which may result in intestinal microbiome disturbance and metabolic disorder in dogs. In conclusion, feeding with SDF did not provide digestive benefits but caused some stress and posed a potential threat to the intestinal health of dogs. Thus, SDF is not recommended in the feeding of dogs.

## 1. Introduction

Dogs serve as great companions, possibly due to their remarkable social cognitive abilities [1]. Owning a dog can provide emotional benefits and improve one’s physical and mental health [2,3,4]. Dog owners care deeply for their pets and treat them as family members, which is evidenced by the increased pet-related expenditure on things such as pet food, veterinary service, training, and pet-sitting [5]. Therefore, maintaining health and expanding the lifespan of dogs could be of significant interest to owners.

The gut is now considered the largest endocrine organ in mammals, and intestinal health can be a top priority in pet health. Gastrointestinal (GI) health not only refers to the aspects of digestion and absorption of nutrients, but also includes the intestinal microbiota, fermentation of product, and metabolite composition [6]. The gut microbiota is a complex ecosystem that has an impact on many areas of the host’s health, including physiology, behavior, and fitness [7]. GI bacteria can influence host physiology and metabolism through direct contact, and also indirectly by way of microbiota-derived metabolites [8,9]. For instance, the GI microbiota aids food digestion and the production of metabolites, including vitamins and short-chain fatty acids (SCFAs) [10]. When there is an imbalance in the gut microbiota, it either helps or hinders intestinal immunity [11]. On the other hand, gut microbiota and metabolites can be affected by different diets in healthy dogs and cats [12,13,14,15]. Therefore, the composition of the microbiota community and fecal metabolites are critical measures for better evaluation and understanding of the gastrointestinal health of dogs.

In the case of young animals, some pet food companies suggest that puffed foods should be softened before use in feeding. However, these statements lack rigorous experiment-based evidence. There has been some research in recent years on the effects of various diets, including fresh and raw meat diets on the apparent total tract digestibility (ATTD), microbiota, and metabolome of the canine gastrointestinal tract [16,17,18]. However, as we know, no study has been carried out on the effect of water-softened dry food (SDF) on dog’s health. Thus, the objective of this study was to evaluate whether feeding with SDF is beneficial to dogs. The effects of a three-week consumption of water-SDF on healthy dogs were examined in the current study by determining the ATTD of macronutrients, immune response, stress hormones, heat shock protein-70 (HSP-70), and fecal characteristics, microbiota, and metabolites.

## 2. Materials and Methods

### 2.1. Animals and Housing

Protocols for all experiments were approved by the Experimental Animal Ethics Committee of South China Agricultural University (protocol code 2021e028). This study included twenty beagle dogs, and their information was shown in Table 1. Subjects were all housed individually kennels measured 1.2 × 1.0 × 1.1 m in an enclosed room with constant temperature (24 ± 1 °C), relative humidity (70 ± 5%) and light control conditions (12 h dark–light cycle, light: 06:00–18:00) in the building of Experimental Animal Centre of South China Agricultural University. All dogs can interact with other dogs and staff daily and were provided opportunities to go out and play at least once a week.

### 2.2. Diets and Feeding

The basal diet used in this experiment was normal commercial dog food (Ramical, Guangzhou, China). The experimental dogs were fed 200 g every day (twice—at 09:00 and 17:00) to meet their energy requirements, which is calculated approximately according to the standard of the National Research Council (NRC, 2000). During the experiment, dogs had free access to clean water at any time through the water nipple. Table 2 showed the nutrient composition of the diet used in this experiment.

### 2.3. Experimental Design

In this study, a total of 20 beagle dogs were chosen. All animals were physically examined at the vet clinic before the start of the trial; vaccination and deworming were completed two months before the experiment. Following an acclimation period of one month when all dogs were fed the basal diet, they were divided into two groups according to their gender and body weight (BW) using a completely randomized block design. One group continually fed the same DF as in the acclimation period (DF, n = 10, male:female = 4:6), and the other group fed SDF was made by the mixture of 1.5 mL purified water with 1 g of dry food (SDF, n = 10, male:female = 4:6). The water used in this test was filtered through a filter to ensure quality. Additionally, the water/dry food ratio was determined to ensure the pet food was neither too tough nor too soft. The whole trial lasted for 21 days. BW and body condition score (BCS) were recorded every ten days, and fecal score (FS) was recorded every day and evaluated as described previously [19]. Fresh fecal samples and blood samples were collected on the last day of the experiment period, and the total fecal collection was performed in the last 3 days.

### 2.4. Chemical Analysis and Digestibility Measurements

Digestibility was measured by the total fecal collection method. A mass of 200 g dry food of basal diet was collected every 10 days, and all feces was collected and 10% HCl was added to the nitrogen fixation for the last three consecutive days in this study. All fecal samples were oven-dried for 48 h (60 °C), then diet and dry fecal samples were ground in a grinder with a 1 mm screen for further chemical analysis. Dry matter (DM) and organic matter (OM) of the diet and feces were determined according to Association of Official Analytical Chemists (AOAC) (2000; method 950.46 and 942.05). A fatty analyzer (FT640, Grand Analytical Instrument Co., Ltd., Guangzhou, China) was used to determine the acid-hydrolyzed fat level (AHF) of the diet and feces (AOAC, 2000; method 954.01). An automatic fiber analyzer (FIBRETHERM FT12, C. Gerhardt GmbH & Co. KG, Königswinter, Germany) was used to analyze the total dietary fiber (TDF) content of the diet and feces (AOAC, 2000; method 962.09). Crude protein (CP) was determined using AOAC with the semi-automatic Kjeldahl apparatus (VAPODEST 200, C. Gerhardt GmbH & Co. KG, Germany). Besides, ATTD values were calculated using the equation as follows: [nutrient intake (g/d, DM basis) − fecal output (g/d, DM basis)]/nutrient intake (g/d, DM basis) × 100.

### 2.5. Blood Sample Collection and Analysis

On day 21, after overnight fasting, 5 mL of blood was taken via forelimb venipuncture. The samples were then clotted for 30 min and centrifugated for 15 min to separate the supernatant for further analyses. Commercial canine enzyme-linked immunosorbent assay (ELISA) kits (MEIMIAN, Jiangsu Meimian Industrial Co., Ltd., Yancheng, China) were used to measure the serum cortisol (COR), adrenocorticotropin hormone (ACTH), glucocorticoids (GC), HSP-70, immunoglobulin G (IgG), tumor necrosis factor-alpha (TNF-α), interferon-γ (INF-γ), interleukin-6 (IL-6), interleukin-4 (IL-4), and interleukin-2 (IL-2).

### 2.6. Microbial Analyses

#### 2.6.1. DNA Extraction, Amplification, and Sequencing

Total genomic DNA was isolated from fresh fecal samples using the cetyltrimethylammonium bromide method. DNA concentration and purity were monitored on a 1% agarose gel. The DNA was diluted to 1 ng/μL with sterile water based on the concentration. The 16S rRNA genes for 16S V3-V4 were amplified using primers 341F (5′-CCTAYGGGRBGCASCAG-3′) and 806R (5′-GGACTACNNGGGGTATCTAAT-3′) with barcodes. For all PCR reactions, a volume of 15 µL Phusion High Fidelity PCR Master Mix (New England Biolabs) was used, along with 2 µM of forward and reverse primers and nearly 10 ng of template DNA. The thermal cycling process was set as follows: initial denaturation for 1 min (98 °C), denaturation for 10 s (98 °C), annealing for 30 s (50 °C), and extension twice for 30 s and 5 min, respectively (72 °C). The PCR products were added with the same volume of (1X) loading buffer (including SYB Green) in an equal density ratio which was subsequently analyzed by electrophoretic manipulation on a 2% agarose gel. Using the Qiagen Gel Extraction Kit, the PCR product combination was then purified (Qiagen, Hilden, Germany). The TruSeq DNA PCR-Free Sample Preparation Kit (Illumina, San Diego, CA, USA) was used to build sequencing libraries, and index codes were then added. A Qubit@ 2.0 fluorometer (Thermo Scientific, Waltham, MA, USA) and an Agilent Bioanalyzer 2100 system were used to evaluate the library’s quality. Finally, the Illumina NovaSeq technology was used to generate final paired-end reads of 250 bp.

#### 2.6.2. Bioinformatics Analysis

Pairs of reads were combined using FLASH (V1.2.7, http://ccb.jhu.edu/software/FLASH/, accessed on 10 June 2022) [20], and the spliced sequence was referred to as the original label. The raw labels were quality filtered according to the quality control process of QIIME (V1.9.1, http://qiime.org/scripts/split_libraries_fastq.html, accessed on 10 June 2022) [21], resulting in high-quality clean labels [22]. Chimeric sequences were detected by comparing the tags with a reference database (Silva database, https://www.arb-silva.de/, accessed on 10 June 2022) using the UCHIME algorithm (UCHIME http://www.drive5.com/usearch/manual/uchime_algo.html, accessed on 10 June 2022) [23]. Then, we removed the chimeric sequences [24] and finally obtained the valid tags.

We used Uparse software (Uparse v7.0.1001, http://drive5.com/uparse/, accessed on 10 June 2022) (Edgar, 2013) to perform sequence analysis. Same OTU were defined as those sequences with ≥97% similarity. For each representative sequence, the classification information was annotated according to the Mothur algorithm and using the Silva database (http://www.arb-silva.de/, accessed on 10 June 2022) [25]. We used MUSCLE software Multiple to performing sequence comparisons (version 3.8.31, http://www.drive5.com/muscle/, accessed on 10 June 2022) [26] to investigate the phylogenetic relationships of different OTUs. Abundance information of OTUs was normalized using sequence number criteria corresponding to the least sequenced sample for further analyses of both alpha-diversity and beta-diversity. Alpha-diversity was measured by six indices, including Observed_species, Chao1, Shannon, Simpson, Ace, and Goods_coverage. All these indices in our samples were calculated using QIIME (version 1.7.0) and displayed using R software (version 2.15.3).

### 2.7. Fecal Short-Chain Fatty Acids and Branch-Chain Fatty Acids Analyses

The gas chromatograph (Shimadzu, Kyoto, Japan) coupled to a flame ionization detector and chromatographic separation capillary column (DB-FFAP, 30 m × 0.25 mm × 0.25 μm) was used in this study to analyze the concentrations of fecal SCFAs (acetic acid, butyric acid, and propionic acid) and branch-chain fatty acids (BCFAs) (isovaleric acid, isobutyric acid, and valeric acid). The instrument parameters and sample processing procedures of extraction of fecal volatiles referred to our previous study [27].

### 2.8. Untargeted Fecal and Serum Metabolomics Analysis

The sample processing procedures of extraction of fecal and serum metabolites referred to our previous study [27]. UPLC-Orbitrap-MS/MS analysis method was used as in a previous study [28], only with slight alterations. Thermo Fisher Scientific’s compound Discover 2.1 data analysis tool was used to scan the mzCloud and mzVault libraries for metabolites. Performing multivariate analysis, MetaboAnalyst 5.0 (https://www.metaboanalyst.ca, accessed on 10 June 2022) was used. In this study, principal component analysis (PCA) and pathway enrichment analyses were performed on MetaboAnalyst 5.0.

### 2.9. Statistical Analysis

SPSS 26.0 was used for statistical analysis, and GraphPad Prism 8.0.3 software was used for graphical presentation. Student’s *t* tests were employed to compare two groups. The mean ± standard error (SE) was used to express all data. Significance was set at *p* < 0.05, a trend was defined by 0.05 < *p* < 0.10. Selecting the metabolites whose adjusted *p* value < 0.05 (calculated by Student’s *t* test) and the variable importance in the projection (VIP) >2 to obtain the differential metabolites.

## 3. Results

### 3.1. Effects of SDF on Body Weight, Body Condition Score, and Fecal Score

As shown in Table 3, there was no difference in BW or BCS between the DF and SDF groups throughout the trial period (*p* > 0.05). The total fecal score (TFS) of the SDF group was markedly higher than that of the DF group (*p* < 0.05). The total soft feces ratio (TFSR) of the DF and SDF groups were both 0.48%. The total diarrhea ratio (TDR) of the SDF group was 0.95%, which was 2.38% higher than TDR of the DF group (3.33%).

### 3.2. Effects of Soften Dry Food on Apparent Total Tract Macronutrient Digestibility

As shown in Table 4, fecal output, ATTD of DM, OM, CP, AHF, and TDF all showed no significant difference between the two groups (*p* > 0.05).

### 3.3. Effects of SDF on Serum Hormones, Heat Shock Protein-70, IgG, and Inflammatory Factors

As shown in Table 5, the SDF group tended to have higher IL-2 (*p* = 0.062) and HSP-70 (*p* = 0.097) levels than the DF group. The SDF group also had a higher level of COR compared with the DF group (*p* < 0.05). Besides, there was no difference shown in IgG, IFN-γ, IL-6, IL-4, TNF-α, GC, and ACTH of the two groups (*p* > 0.05).

### 3.4. Effects of SDF on Fecal Microbiota

As shown in Figure 1A, alpha diversity analyses of all samples were performed. Chao1 and Ace represent species richness, while the Shannon and Simpson indices are indicative of a diverse microbial population. We can see that the fecal microbial communities of the SDF group had more Observed_species, higher Ace and Chao1 indices levels than that of the DF group, whereas the fecal microbial of the SDF group had lower Goods_coverage over that of the DF group (*p* < 0.05). Besides, fecal samples were evaluated for beta diversity to analyze the structural difference of microbial community (Figure 1B). There was no separation between the DF and SDF groups in PCoA of weighted UniFrac distances (*p* > 0.05). However, PCoA of unweighted UniFrac distances revealed distinct separation between two groups (*p* < 0.05), which showed that SDF could influence the gut microbiota composition and structure in dogs.

The top 6 abundant phyla included Firmicutes, Fusobacteria, Bacteroidetes, Actinobacteria, Actinobacteriota, and Proteobacteria (Figure 1C). Additionally, the relative abundances at the family level indicated that the top 8 abundant families included Lactobacillaceae, Peptostreptococcaceae, Erysipelotrichaceae, Fusobacteriaceae, Ruminococcaceae, Streptococcaceae, Lachnospiraceae, and Muribaculaceae (Figure 1C). Finally, the top 10 abundant genera were Lactobacillus, Peptoclostridium, Fusobacterium, Allobaculum, Cetobacterium, Faecalibacterium, Streptoboccaceae, Blautia, Romboutsia, and Bifidobacterium (Figure 1C).

To select the differential bacteria, differential abundant taxa were confirmed by LEfSe analysis (LDA > 3.0). Proteobacteria and Acidobacteriota were enriched in the SDF group at the phylum level. At the family level, Proteobacteria, Enterobacteriaceae, Enterococcaceae, and Sphingomonadaies were enriched in the SDF group, and Ruminococcaceae was enriched in the DF group. Furthermore, at a genus level, Streptococcus, Enterococcus, and Escherichia_Shigella were enriched in the SDF group, and Faecalibacterium was enriched in the DF group (Figure 1D).

### 3.5. Effects of SDF on Fecal Short-Chain Fatty Acids and Branch-Chain Fatty Acids

As shown in Table 6, no significant difference occurred in the concentrations of fecal total SCFAs, acetic acid, propionic acid, butyric acid, total BCFAs, valeric acid, isovaleric acid, and isobutyric acid (*p* > 0.05) between the DF and SDF groups.

### 3.6. Effects of SDF on Fecal Metabolome

To investigate the metabolic regulation in the gut of SDF, the fecal metabolites were analyzed using UPLC-Orbitrap-MS/MS. Metabolomics analysis from all fecal samples identified 249 metabolites. PCA of the fecal metabolomics shows that SDF influenced the fecal metabolites in the gut of dogs (Figure 2A). Furthermore, the Orthogonal Partial Least-Squares-Discriminant Analysis (ortho PLS-DA) for fecal metabolomics in two groups shows significant separation (Figure 2B). This result confirms the effects of SDF on fecal metabolites.

Twenty-four fecal metabolites were significantly different between the DF and SDF groups (Table 7). Dogs fed SDF had higher contents of isoquinoline, N-Acetylmuramic acid, riboflavin, guanosine, glutamylglutamine, tigilc acid, 5-Aminopentanoic acid, taurine, L-Glutamine, guanine and deoxyadenosine, and lower contents of 16-Hydroxy hexadecenoic acid, stearoylethanolamide, tryptophanol, N, N-Dimethylsphingosine, 4-Dodecylbenzenesulfonic acid Na salt, oleamide, anandamide, sebacic aced, 2-Piperidinone, carnosine, palmitic amide, adenosine, and fluoxymesteron. The enrichment analysis platform of Metaboanalyst 5.0 was adopted to explore the differential metabolic pathways (Figure 2C). Among these, purine metabolism (nucleotide metabolism) and riboflavin metabolism (metabolism of cofactors and vitamins) were significantly influenced. The specific fecal metabolites enriched in the differential pathways were showed in the bar plot (Figure 2D).

### 3.7. Effects of SDF on Serum Metabolome

To further investigate the effects of an SDF diet on metabolic regulation in dogs, we used UPLC-Orbitrap-MS/MS to analyze the serum metabolites. Metabolomics analysis from all serum samples identified 126 metabolites. PCA of the serum metabolomics showed no significant separation between two groups (Figure 3A). However, the ortho PLS-DA for serum metabolomics in the two groups shows significant separation (Figure 3B). This result shows that the SDF influenced the serum metabolites in dogs.

We selected the differential metabolites with VIP > 1.5 and *p* < 0.05, and 13 serum metabolites were significantly different between the DF and SDF groups (Table 8). Dogs fed SDF had higher contents of 3-Hydroxycapric acid, 3-indolebutyric acid, cis-4-hydroxy-D-proline, isoquinoline, alpha-hydroxyisobutyric acid, and L-glutamine, lower contents of N-acetylornithine, fluoren-9-one, 3-cresotinic acid, 3,4-dihydroxyhydrocinnamic acid, N-acetyl-L-phenylalanine, and indole-3-propionic acid. The enrichment analysis platform of Metaboanalyst 5.0 was adopted to explore the differential metabolic pathways (Figure 3C). Among these, arginine and proline metabolism were significantly influenced. Furthermore, the specific serum metabolites enriched in the differential pathways were showed in the bar plot (Figure 3D).

### 3.8. Correlation between Differential Metabolites and Fecal Bacteria at the Genus Level

Correlation between differential metabolites (fecal and serum) and fecal bacteria (at the genus level) is displayed in the heatmap (Figure 4). As we can see that Aeromounas were negatively associated with fluoxymesterone and fluoren-9-one, Dubosiella were negatively associated with fluxoymesterone, 4-Dodecylbenzenesulfonic acid Na salt, palmitic amide, 3-cresotinic acid and N-acetyo-L-phenylalanine. Besides, Collinsella had a positive association with isoquinoline, and Escherichia-Shigella had a positive association with serum cortisol, 3-indolebutyric acid, and isoquinoline.

## 4. Discussion

With scientific and healthy concepts being introduced more in pet feeding strategies, pet owners are paying extra attention to the diet quality and intestinal health of their pets. Although some pet food companies recommend that feeding pets with SDF is better for their health, there is no rigorous experiment-based evidence to suggest this. Therefore, we investigated the effects of SDF on ATTD, immune response, stress indicators, and fecal characteristics, microbiota, and metabolites in dogs.

Nutrient digestibility is an important measurement for evaluating the quality of a diet for dogs [29]. Many factors, including dietary ingredients, processing mode, and animal’s physiology state can affect the nutrient digestibility, while fecal output may also be influenced by food intake, the chemical composition of the diet, physiological state, and nutrient digestibility of animals [30]. Higher nutrient digestibility usually results in lower fecal output. This can correspond to the results of the current study. Under the premise of daily intake of the same amounts of foods, there were no significant differences in fecal output and ATTD. Additionally, SDF had no effect on nutrient digestibility in dogs. In addition, nutrient digestibility, fiber content, DM intake, and fat tolerance can affect the fecal quality [31,32]. Even though the basal diet was the same for both groups in the present study, SDF resulted in softer stools and a higher diarrhea ratio than DF. This may be caused by the disturbance of the intestinal bacteria.

In this study, serum COR of the SDF group was significantly higher than that of the DF group on day 21. An increase in COR is commonly observed in stress response where internal and external stimulus activates the hypothalamic-pituitary-adrenal axis and triggers a cascade of hormonal release [33,34]. Besides, we found that serum COR level was positively associated with *Escherichia-Shigella*, indicating that the increase in *Escherichia-Shigella* may cause stress to dogs fed with SDF, thereby inducing an increase in th serum COR level. Additionally, the SDF group tended to have higher serum IL-2 and HSP-70 levels than the DF group. IL-2 is the major regulatory hormone of the immune system [35]. Heat shock proteins are induced by a wide variety of stressors and have broad cytoprotective functions; in particular, HSP-70 plays a vital role in cellular protection [36,37,38]. This may suggest that feeding with SDF may exert stress and have a tendency to cause inflammation in dogs.

In addition to the signs of gastrointestinal health (e.g., fecal state, diarrhea ratio, and volume) that are noticeable to pet owners, the abundance and activity of gut microbiota are critical to the long-term health [39,40,41]. Different diets can regulate the gut microbiota [12]. In our study, the results of alpha diversity in fecal microbial communities revealed significant differences between the two groups on day 21. Changes in the microbial communities occurred in the SDF group, which had more Observed_species, higher Ace and Chao1 indices, and lower Goods_coverage than the DF group. Moreover, a significant separation occurred between two groups in PCoA analysis based on unweighted UniFrac distances. In short, feeding with SDF could increase the species richness, the diversity of the gut microbiota and change the intestinal microbiome structure in dogs. The most abundant phyla found in the two groups were Firmicutes, Fusobacteria, Bacteroidetes, Actinobacteria, and Proteobacteria, which are similar to the previous studies [16,17,42]. The results of LEfSe showed that feeding dogs with SDF significantly changed with phyla Proteobacteria and Acidobacteriota, families Streptococcaceae, Enterococcaceae, Bacillaceae, Sphingomonadaceae, and Ruminococcaceae. Besides, genus *Streptococcus*, *Collinsella*, *Dubosiella*, *Escherichia_Shigella*, and *Enterococcus* were proved to be significantly enriched in the SDF group. Among these bacteria, *Streptococcus*, *Escherichia_Shigella*, and *Enterococcus* are known as pathogen bacterial associated with idiopathic small intestinal irritable bowel disease [40,43], and a previous study showed that *Escherichia_Shigella* had a strong negative correlation with *Faecalibacterium* [44], indicating that SDF caused the growth of some pathogenic bacteria. Similar to this study, our study results also showed that *Escherichia_Shigella* was enriched in the SDF group and *Faecalibacterium* was enriched in the DF group. *Collinsella*, the major taxon of the Coriobacteriaceae family, has a close correlation to insulin circulation [45]. *Dubosiella* has been recently identified as a novel member of the family Erysipeotrichaceae which thrives in such conditions, and a study showed it was enriched in the gut of obese LKO mice fed a high-fat diet [46,47]. The exact functions of *Dubosiella* are not clear, but a study showed the abundances of *Dubosiella* increased in DSS-induced mice [48]. Thus, we can speculate that the increase in fecal *Collinsella* and *Dubosiella* in the SDF group may result in some potential inflammation in the present study. In conclusion, changes in bacterial phylogeny observed are not always indicative of functional changes. However, according to the results of previous studies, most of the increased bacteria genera in the SDF group are potential pathogenic bacteria from previous study; that is, SDF may lead to an increase to the intestinal pathogens. Moisture content is an important factor in the growth of bacteria in pet food, and SDF may increase the risk.

An abundance of gut microbial-derived metabolites exists in the digestive tract [49]. In this study, there was no difference in fecal SCFAs and BCFAs between the SDF and DF groups. However, fecal metabolites derived from microbiome activity are usually secreted into the intestine and transferred to the circulatory system through the intestinal barrier, serving as very important regulators of host metabolism [9,50]. Metabolomics helps to identify the key metabolites in different diets. Our study used UPLC-Orbitrap-MS/MS to analyze and investigate the changes in fecal and serum metabolites in dogs with 21 days of SDF feeding.

The results of fecal metabolites analyses showed that SDF significantly influenced the purine metabolism and riboflavin metabolism in dogs, and that there were five differential metabolites (guanine, L-glutamine, guanosine, adenosine, and ribofalvin) enriched in these two differential pathways. A study indicated that the release of guanine increases following hypoxia/hypoglycemia and permits astrocytes to exert neurotrophic effects [51]. Guanosine may have a function in stress response, biofilm development, and cellular damage protection [52,53,54]. A study reported that L-glutamine is an abundant nitrogen source in host serum and cells and serves as an environmental indicator and inducer of virulent gene expression [55]. Adenosine is an inhibitory modulator of neuronal activity, with potent inhibitory effects on synaptic activity. In a study, extracellular adenosine levels increase dramatically in the brain via hydrolysis of adenosine triphosphate during conditions of high energy demand [56]. Another study demonstrated that adenosine interferes with the production and release of various inflammatory mediators, and it activates cellular antioxidant defense systems, thus providing protective effects at multiple levels in the pathogenesis of ischemia and reperfusion [57]. Except for the above metabolites, some other differential metabolites are also worth noting. Taurine has been linked to antioxidant functions and defense against oxygen free radicals [58], as well as apoptosis, inflammation, cell death, and increasing NO production in endothelial cells [59,60]. Taurine can alleviate oxidative stress either directly by converting superoxides to taurine chloramine [61], or indirectly through a variety of processes, including the renin-angiotensin system [62]. In conclusion, the higher contents of guanosine and taurine in the SDF group also seem to indicate oxidative stress. In both in vitro and in vivo studies, oleamide has been shown to have anti-inflammatory and anti-allergenic properties [63,64,65]. Conversely, a study showed that oleamide and palmitic amide may significantly and positively correlate with activities circulating inflammatory markers, including TNF-α, hypersensitive-c-reactive-protein, and Lipoprotein-associated phospholipaseA2 activities [66]. While in this study, the lower contents of oleamide and palmitic amide may increase inflammation in dogs fed with SDF.

Further serum metabolites analyses showed that SDF significantly influenced the arginine and proline metabolism, and that cis-4-Hydroxy-D-proline was the differential metabolite in the differential pathway. However, the function and mechanism of cis-4-Hydroxy-D-proline in the serum are still unknown. Besides, high levels of 3-indolebutyric acid (IBA) and isoquinoline were observed in the SDF group. IBA, the naturally occurring metabolite of tryptophan, a study shows that it was the metabolic product of Clostridia species [67]. In another study, IBA was proven to co-occur with the incidence of inflammatory bowel syndromes in schizophrenic patients [68], and a potential mechanism for IBA to control inflammation is competitive inhibition of phospholipase A2 [69]. Isoquinoline was a type of cardiac glycosides and potent apoptosis inducers. Its derivatives are widely presented in many plants and foodstuffs, and readily cross the blood–brain barrier. In addition, isoquinoline derivatives are oxidized by monoamine oxidases to produce isoquinolinium cations with the concomitant generation of reactive oxygen species [70]. According to correlation analysis, we found that 3-indolebutyric acid and isoquinoline were positively associated with *Escherichia-Shigella*. Therefore, we suggest that 3-indolebutyric acid and isoquinoline could serve as potential biomarkers of the SDF stress response. However, the relationship between intestinal microbiome and metabolites needs further investigation.

These results could adequately explain the result of the increased diarrhea ratio and FS in dogs, and the increase in the pathogens caused some stress to dogs, thereby leading to the elevated serum IL-2, COR, and HSP-70. Based on the changes of specific bacteria and different metabolites, we speculated that bacteria may breed in the process of water-SDF, which leads to the increase in TFS and TDR, as well as abnormal fecal shape. However, although we observed that the contents of some bacteria and metabolites were changed and had some correlation, the exact biological functions and the underlying mechanisms in dogs still need further exploration.

## 5. Conclusions

We found that feeding with SDF did not influence the ATTD, fecal SCFAs and BCFAs in dogs. However, this feeding method increased the diarrhea ratio, serum stress indicators levels, and risk of inflammation. Although feeding with SDF increased the diversity of the intestinal microbiotas in dogs, it caused a significant increase in some pathogenic bacteria. Meanwhile, fecal metabolomics further revealed that feeding with SDF can induce metabolic disorders in dogs. In conclusion, feeding with SDF did not provide digestive benefits, but posed some stress and a potential threat to the intestinal health of dogs. Thus, SDF is not recommended in dogs.

## Figures and Tables

**Figure 1 metabolites-12-01124-f001:**
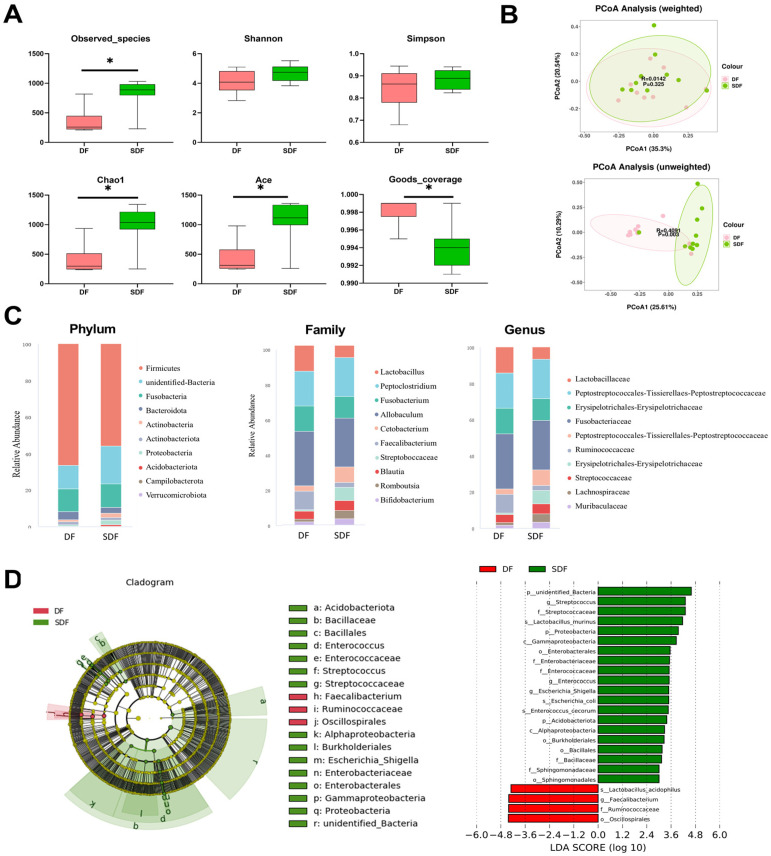
Alpha diversity indices (Observed_species, Shannon, Simpson, Chao1, Ace and Goods_coverage) of fecal microbial communities of the DF and SDF groups on day 21 (**A**). The symbol (*) indicates statistically significant differences between two groups. Principal Coordinate Analysis (PCoA) based on weighted and unweighted UniFrac distances of fecal microbial communities of the DF and SDF groups on day 21 (**B**). Predominant fecal microbial communities and different bacteria at the phylum, family, and genus levels of the DF and SDF groups on day 21 (**C**). LEfSe analysis identified gut bacterial biomarkers of the DF and SDF groups on day 21 (**D**). The histogram of the LDA scores presents species (biomarker) whose abundance showed significant differences between different groups. The LDA score represents the effect size. In the cladogram, circles radiating from the inner side to outer side represent taxonomic level from phylum to genus (species). Each circle’s diameter is proportional to the taxon’s relative abundance. Yellow nodes refer to the bacteria that contributed a lot in the DF group and blue nodes refer to the bacteria dominant in the SDF group. DF = dry food; SDF = softened dry food.

**Figure 2 metabolites-12-01124-f002:**
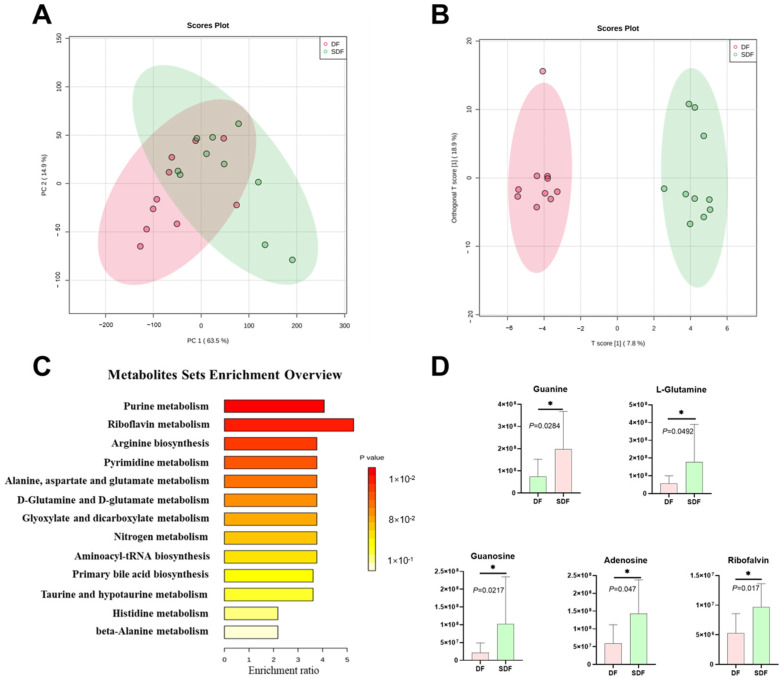
Principal components analysis (PCA) for fecal metabolomics data of dogs fed DF and SDF on day 21 (**A**). Orthogonal Partial Least-Squares-Discriminant Analysis (orthoPLS-DA) for fecal metabolomics data of dogs fed DF and SDF on day 21 (**B**). Enrichment bar plot based on the fecal differential metabolites of dogs fed DF and SDF on day 21 (**C**). Bar plot based on the fecal differential metabolites which were enriched in the differential metabolite pathways between two groups (**D**). The symbol (*) indicates the difference between the dry food (DF) and softened dry food (SDF) groups.

**Figure 3 metabolites-12-01124-f003:**
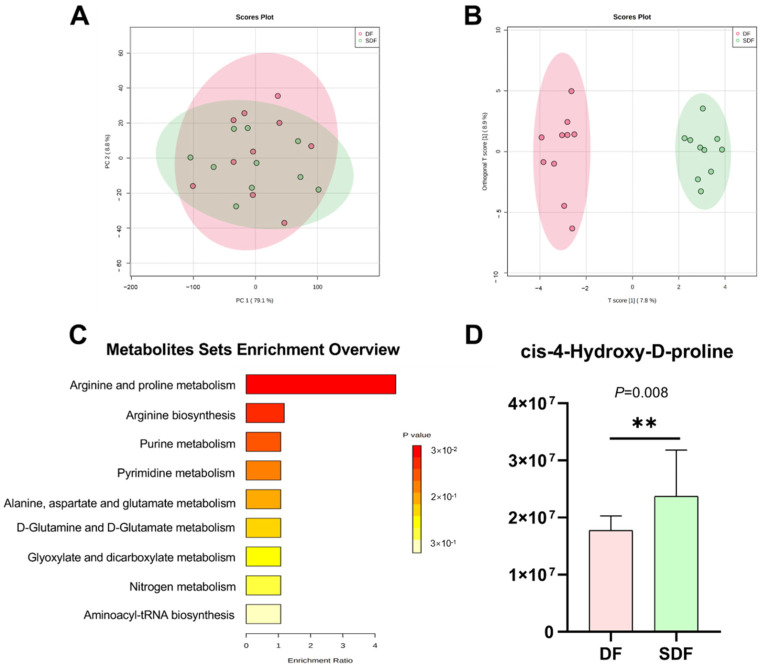
Principal components analysis (PCA) for serum metabolomics data of dogs fed dry food (DF) and softened dry food (SDF) on day 21 (**A**). Orthogonal Partial Least-Squares-Discriminant Analysis (orthoPLS-DA) for serum metabolomics data of dogs fed dry food (DF) and softened dry food (SDF) on day 21 (**B**). Enrichment bar plot based on the serum differential metabolites of dogs fed dry food (DF) and softened dry food (SDF) on day 21 (**C**). Bar plot based on the serum differential metabolites which were enriched in the differential metabolite pathways between two groups (**D**). The symbol (**) indicates the difference between the dry food (DF) and softened dry food (SDF) groups.

**Figure 4 metabolites-12-01124-f004:**
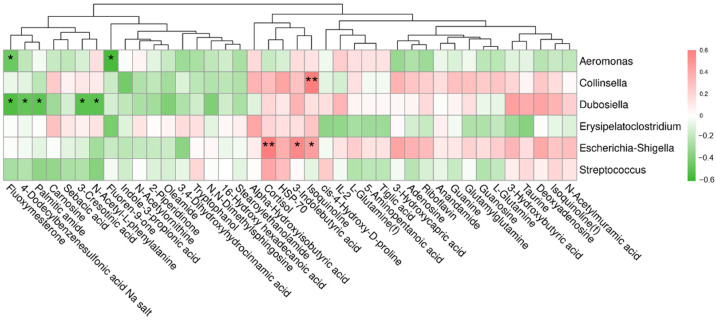
Correlation heatmap between differential metabolites and differential genera in the dogs on day 21. The symbol (*) indicates a significant correlation between serum metabolites and fecal bacteria (* *p* < 0.05, ** *p* < 0.01). Pink color indicates a positive correlation, and green color indicates a negative correlation.

**Table 1 metabolites-12-01124-t001:** Detailed information of dogs in this study.

Group ^1^	N (Male: Female)	Age (M)	Weight (kg)
DF	10 (4:6)	5.38 ± 0.35	7.33 ± 0.12
SDF	10 (4:6)	5.62 ± 0.29	7.25 ± 0.18

^1^ DF: dry food, SDF: water-softened dry food.

**Table 2 metabolites-12-01124-t002:** The chemical composition of the basal diet.

Items ^1^	Basal diet ^2^
DM (%)	91.19
OM (% DM)	91.57
CP (% DM)	27.69
AHF (% DM)	11.21
TDF (% DM)	3.63

^1^ DM: dry matter; OM: organic matter; CP: crude protein; AHF: acid-hydrolyzed fat; TDF: total dietary fiber. ^2^ Extruded diet: corn flour, flour, sweet potato, chicken powder, meat and bone meal, soybean meal, beef meal, calcium hydrophosphate, calcium chloride, lysine, methionine, vitamin A, vitamin D, vitamin E, copper sulfate, ferrous sulfate, zinc sulfate, and manganese sulfate.

**Table 3 metabolites-12-01124-t003:** Body weight, body condition score, and fecal score in dogs fed DF and SDF on days 11 and 21.

Item ^1^	DF ^2^	SDF ^3^	*p* Value
BW (kg)	8.01 ± 0.14	7.87 ± 0.18	0.540
BCS	5.05 ± 0.17	4.65 ± 0.11	0.660
TFS	2.55 ± 0.14	2.62 ± 0.25	0.028
TSFR (%)	0.48	0.48	-
TDR (%)	0.95	3.33	-

^1^ BW: body weight; BCS: body condition score; all the test dogs were weighed and BCS were recorded with a nine-point scale on days 1, 11, and 21 before morning feeding. TFS: total fecal score. All samples were scored in 0.5 units increments using the following scale: 1 = hard, dry pellets, small hard masses; 2 = hard, stool, remains firm and soft; 3 = soft, formed, and moist stool, retains shape; 4 = soft, unformed stool, assumes shape of container; 5 = watery, liquid that can be poured; in which, 1 ≤ FS < 2 constipation, 2 ≤ FS ≤ 3 normal, 3 < FS < 4 soft stool, 4 ≤ FS ≤ 5 diarrhea. TSFR: total soft feces ratio. Ratio of the number of soft feces to the total number of fecal scores. TDR: total diarrhea ratio. Ratio of the number of diarrhea to the total number of fecal scores. ^2^ DF: dry food. ^3^ SDF: water-softened dry food.

**Table 4 metabolites-12-01124-t004:** Food intake, fecal characteristics, and apparent total tract macronutrient digestibility in dogs fed DF and SDF.

Item	DF ^1^	SDF ^2^	*p* Value
Food intake ^3^, g/d (DM basis)	184.00	184.00	-
Fecal output, g/d (as is)	91.07 ± 4.78	89.00 ± 1.81	0.693
Fecal output, g/d (DM basis)	24.90 ± 1.27	23.82 ± 1.73	0.628
Digestibility, %			
Dry matter	88.55 ± 0.01	89.04 ± 0.01	0.628
%DM basis			
Organic matter	81.28 ± 0.26	82.39 ± 0.10	0.693
Crude protein	83.88 ± 0.01	85.52 ± 0.01	0.301
Acid hydrolyzation fat	94.83 ± 0.01	95.84 ± 0.01	0.104
Total dietary fiber	70.73 ± 0.02	66.82 ± 0.01	0.202

^1^ DF: dry food. ^2^ SDF: water-softened dry food. ^3^ All dogs were fed a restricted diet during the trial and all dogs had no leftovers at each meal.

**Table 5 metabolites-12-01124-t005:** Serum IgG, inflammatory factors, hormones, and HSP-70 in dogs fed DF and SDF on day 21.

Item ^1^	DF ^2^	SDF ^3^	*p* Value
IgG	258.88 ± 16.04	227.38 ± 14.90	0.167
IFN-γ	42.10 ± 1.39	45.23 ± 1.23	0.108
IL-6	541.77 ± 20.15	567.00 ± 21.69	0.405
IL-4	163.72 ± 14.20	146.66 ± 9.65	0.334
IL-2	254.86 ± 11.54	287.23 ± 11.42	0.062
TNF-α	77.08 ± 4.15	77.12 ± 4.69	0.995
GC	321.91 ± 13.13	318.80 ± 9.23	0.887
ACTH	67.23 ± 3.31	68.10 ± 2.45	0.836
COR	394.24 ± 13.13	438.50 ± 9.23	0.014
HSP-70	334.50 ± 18.05	376.07 ± 15.42	0.097

^1^ IgG = immunoglobulin G; IFN-γ = interleukin-γ; IL-6 = interleukin-6; IL-4 = interleukin-4; IL-2 = interleukin-2; ACTH = adreno-cortico-tropic-hormone; TNF-α = tumor necrosis factor; GC = glucocorticoid; COR = cortisol; HSP-70 = heat stress protein 70. ^2^ DF: dry food. ^3^ SDF: water-softened dry food.

**Table 6 metabolites-12-01124-t006:** Fecal SCFAs and BCFAs concentrations in dogs fed DF and SDF.

Item (µg/g)	DF ^1^	SDF ^2^	*p* Value
Acetic acid	1476.31 ± 35.88	1479.08 ± 54.89	0.968
Propionic acid	1079.94 ± 23.41	1063.59 ± 43.24	0.751
Butyric acid	442.42 ± 19.53	443.47 ± 13.93	0.891
Total SCFAs ^3^	2998.66 ± 72.02	2986.13 ± 103.37	0.924
Isobutyric acid	92.21 ± 7.12	91.06 ± 9.95	0.955
Isovaleric acid	142.52 ± 12.76	141.41 ± 14.25	0.550
Valeric acid	19.70 ± 3.26	17.35 ± 2.20	0.891
Total BCFAs ^4^	254.43 ± 20.20	249.82 ± 25.68	0.922

^1^ DF: dry food. ^2^ SDF: softened dry food. ^3^ Total SCFAs: total short-chain fatty acids = acetic acid + propionic acid + butyric acid. ^4^ Total BCFAs: total branch-chain fatty acids = isobutyric acid + isovaleric acid + valeric acid.

**Table 7 metabolites-12-01124-t007:** Fecal differential metabolites were identified according to the standard of VIP > 1.5 and *p* < 0.05 between the DF and SDF groups.

Metabolites	*p* Value	VIP ^1^	KEGG ^2^	Trend (SDF ^3^ vs. DF ^4^)
16-Hydroxy hexadecenoic acid	0.000	2.03	-	Down
Stearoylethanolamide	0.000	1.96	-	Down
Isoquinoline	0.002	2.01	C06323	Up
Tryptophanol	0.002	2.26	C00955	Down
N, N-Dimethylsphingosine	0.002	2.17	C13914	Down
N-Acetylmuramic acid	0.004	1.58	C02713	Up
Riboflavin	0.004	2.28	C00255	Up
4-Dodecylbenzenesulfonic acid Na salt	0.004	2.24	-	Down
Oleamide	0.005	2.03	C19670	Down
Anandamide	0.005	1.63	C11695	Down
Guanosine	0.006	1.98	C00387	Up
Sebacic acid	0.006	2.15	C08277	Down
Glutamylglutamine	0.012	1.92	-	Up
2-Piperidinone	0.020	2.12	-	Down
Carnosine	0.027	2.08	C00386	Down
Tiglic acid	0.027	1.73	C08279	Up
Palmitic amide	0.023	1.82	-	Down
5-Aminopentanoic acid	0.028	1.72	C00431	Up
Taurine	0.029	1.61	C00245	Up
Adenosine	0.034	1.51	C00212	Down
L-Glutamine	0.035	1.56	C00064	Up
Fluoxymesterone	0.036	1.57	-	Down
Guanine	0.039	1.89	C00242	Up
Deoxyadenosine	0.040	1.51	C00559	Up

^1^ VIP: variable importance in the projection. ^2^ KEGG: Kyoto Encyclopedia of Genes and Genomes. ^3^ SDF: water-softened dry food. ^4^ DF: dry food.

**Table 8 metabolites-12-01124-t008:** The serum differential metabolites were identified according to the standard of VIP > 1.5 and *p* < 0.05 between the DF and SDF groups.

Metabolites	*p* Value	VIP ^1^	KEGG ^2^	Trend (SDF ^3^ vs. DF ^4^)
3-Hydroxycapric acid	0.001	2.49	-	Up
N-Acetylornithine	0.004	2.13	C00437	Down
3-Indolebutyric acid	0.006	2.16	C11284	Up
cis-4-Hydroxy-D-proline	0.008	2.02	C03440	Up
3-Hydroxybutyric acid	0.011	1.94	C01089	Up
Isoquinoline	0.011	2.05	C06323	Up
Fluoren-9-one	0.012	1.87	C06712	Down
3-Cresotinic acid	0.022	1.89	C14088	Down
3,4-Dihydroxyhydrocinnamic acid	0.024	1.84	C10447	Down
N-Acetyl-L-phenylalanine	0.026	1.85	C03519	Down
Alpha-Hydroxyisobutyric acid	0.027	1.90	-	Up
L-Glutamine	0.030	1.76	C00064	Up
Indole-3-propionic acid	0.042	1.83	-	Down

^1^ VIP: variable importance in the projection. ^2^ KEGG: Kyoto Encyclopedia of Genes and Genomes. ^3^ SDF: water-softened dry food. ^4^ DF: dry food.

## Data Availability

The datasets presented in this study can be found in online repositories. The names of repository/repositories and accession number(s) can be found below: https://www.ncbi.nlm.nih.gov/sra/PRJNA898496 (accessed on 5 November 2022).

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
