# Peer review of "Effects of Softening Dry Food with Water on Stress Response, Intestinal Microbiome, and Metabolic Profile in Beagle Dogs"

_metabolites, 2022, doi:10.3390/metabo12111124_

Round 1

Reviewer 1 Report

This study aimed to compare feeding dry food (DF) and water-softened dry food (SDF) on stress response, intestinal microbiome, and metabolic profile in dogs. The introduction is sufficient to explain why this study was conduct. Material and methods are well described. Results are interesting and clear. Discussion is well explained the results.

1. some text in Figure 1C, Figure 2A, 2B and Figure 3A, 3B are not clear.

2. Please provide a picture of beagle dogs in 2.1 section

3. Line 530. Please deposit your raw data in a public database

Author Response

Dear reviewer,

We would like to thank you for your careful reading, helpful comments, and constructive suggestions, which has significantly improved the presentation of our manuscript. Here, we submit a revised version of our manuscript entitled “Effects of Softening Dry Food with Water on Stress Response, Intestinal Microbiome, and Metabolic Profile in Beagle Dogs”, which has been modified according to your suggestions.We summarized our responses to each comment from you in the following sections. And the following is a point-to-point response to your comments. 

Comment 1: some text in Figure 1C, Figure 2A, 2B and Figure 3A, 3B are not clear.

Response: Thanks for your suggestion, we have replaced figure1, figure 2, and figure 3 with high quality images in the revised manuscript. Please see lines 266, 312, and 343.

Comment 2: Please provide a picture of beagle dogs in 2.1 section.

Response: Thanks for your carefulness. According to previous companion animals nutrition studies, there were few articles showed pictures of dogs. Therefore, we did not show picture of beagle dogs in paper.

Comment 3: Line 530. Please deposit your raw data in a public database.

Response: Thank for your question. We have uploaded the raw data. The names of repository/repositories and accession number(s) can be found below: https://www.ncbi.nlm.nih.gov/sra/PRJNA898496. Please see line 523-525.

We believe that our responses have well addressed all concerns from you. We hope our revised manuscript can be accepted for publication. And if you have any question about this paper, please don’t hesitate to let us know.

Sincerely yours,

Jinping Deng

Reviewer 2 Report

The manuscript titled "Effects of Softening Dry Food with Water on Stress Response, Intestinal Microbiome, and Metabolic Profile in Beagle Dogs is very interesting and, in my opinion, an effort was made to present the collected data in an interesting and reliable way.

However, the authors could consider a few points, which I present below.

Line 61: Check if there was a translation of the abbreviation after its first use

Line 110: In my opinion, the assay methodologies were presented too laconically and this should be supplemented with more details

Line 175: The gas chromatograph (Shimadzu, Tokyo, Japan)? Or The gas chromatograph (Shimadzu, Kyoto, Japan)? Please check.

Line 290-291: There is no need to provide the same information again regarding the method used.

Furthermore, based on the description in the manuscript, I judge that the welfare of the animals was maintained by the researchers during the experiment.

Author Response

Dear reviewer,

We would like to thank you for your careful reading, helpful comments, and constructive suggestions, which has significantly improved the presentation of our manuscript. Here, we submit a revised version of our manuscript entitled “Effects of Softening Dry Food with Water on Stress Response, Intestinal Microbiome, and Metabolic Profile in Beagle Dogs”, which has been modified according to your suggestions.We summarized our responses to each comment from you in the following sections. And the following is a point-to-point response to your comments. 

Comment 1: Line 61: Check if there was a translation of the abbreviation after its first use.

Response 1: Thank you for correcting the mistake, we replaced “SDF” with “water-softened dry food (SDF)” in red font. Please see line 65.

Comment 2: Line 110: In my opinion, the assay methodologies were presented too laconically and this should be supplemented with more details.

Response 2: Thank you for this valuable feedback. Revised portion are marked in red in the paper, please see line 112-114.

Comment 3: Line 175: The gas chromatograph (Shimadzu, Tokyo, Japan)? Or The gas chromatograph (Shimadzu, Kyoto, Japan)? Please check.

Response 3: Thank you for your question. We checked and confirmed it is gas chromatograph (Shimadzu, Kyoto, Japan). Revised portion are marked in red font. Please see line 178.

Comment 4: Line 290-291: There is no need to provide the same information again regarding the method used.

Response 4: We are so grateful for your kind question. We have removed the same information in red font. Please see line 300-301.

We believe that our responses have well addressed all concerns from you. We hope our revised manuscript can be accepted for publication. And if you have any question about this paper, please don’t hesitate to let us know.

Sincerely yours,

Jinping Deng

Reviewer 3 Report

Regarding MS ‘’ Effects of Softening Dry Food with Water on Stress Response, Intestinal Microbiome, and Metabolic Profile in Beagle Dogs’’ this study is interesting, but I have some comments:

 L17. The trial lasted for 21 days; this is a short period!

L65. Hypothesis is missing

L59-60 Up till now, few studies investigated if feeding SDF is beneficial to the gastrointestinal health of dogs. Please briefly mention some of these studies and what they are found. The authors should describe the knowledge gap; in other words, why they performed this study?

L99. How many replicates per group?

L104. On which basis the authors chose this ratio of 1.5 ml water to g of feed

L126. Number of animals for sampling? Is the design, a completely randomized design, or a completely randomized block design? Please clarify.

 Why did the authors not use marker in the digestibility trial?

Table 3. TSFR (%) and TDR (%), missing statistics.

L515-516. These two sentences should be added to the abstract (end of the abstract).

Author Response

Dear reviewer,

We would like to thank you for your careful reading, helpful comments, and constructive suggestions, which has significantly improved the presentation of our manuscript. Here, we submit a revised version of our manuscript entitled “Effects of Softening Dry Food with Water on Stress Response, Intestinal Microbiome, and Metabolic Profile in Beagle Dogs”, which has been modified according to your suggestions.We summarized our responses to each comment from you in the following sections. And the following is a point-to-point response to your comments. 

Comment 1: L17. The trial lasted for 21 days; this is a short period!

Response 1: Thanks for the comment. Although the whole trial only lasted for 21 days, the data are already able to conclude that feeding SDF did not provide digestive benefits but pose some stress and a potential threat to the intestinal health of dogs. Thus, a 21-day short-term trial is sufficient.

Comment 2: L65. Hypothesis is missing.

Response 2: Thank you for your kind question. Some pet food companies suggest that puffed foods should be softened before feeding for young animals. However, no academic paper has shown its beneficial. Therefore, this study aimed to compare feeding dry food and water-softened dry food on stress response, intestinal microbiome, and metabolic profile in dogs.

Comment 3: L59-60 Up till now, few studies investigated if feeding SDF is beneficial to the gastrointestinal health of dogs. Please briefly mention some of these studies and what they are found. The authors should describe the knowledge gap; in other words, why they performed this study?

Response 3: Thank you for your question. As far as we know, no academic paper has shown that feeding SDF is beneficial to the gastrointestinal health of dogs, and no studies have been conducted to determine whether feeding SDF has any other effects. We have designed and conducted this experiment systematically for the first time.

Comment 4: L99. How many replicates per group?

Response 4: Thank you for your question. There are 10 replicates per group.

Comment 5: L104. On which basis the authors chose this ratio of 1.5 ml water to g of feed.

Response 5: Thank you for your question. We did a comparison before experiment and found that use 1.5 ml water to g of feed softened the food but not loose.

Comment 6: L126. Number of animals for sampling? Is the design, a completely randomized design, or a completely randomized block design? Please clarify.

Response 6: Thank you for your question. We have 20 beagles for sampling, and the beagle dogs were randomly allotted to two groups according to their gender and body weight. Thus, this experiment used a completely randomized block design. And we have added it to the abstract and experiment design in blue font, please see lines 18-19 and 100-104.

Comment 7: Why did the authors not use marker in the digestibility trial?

Response 7: Thanks for your suggestion. We used metabolic cages in this trial, which allowed us to collect total feces from each dog, so we did not use marker in the digestibility trial.

Comment 8: Table 3. TSFR (%) and TDR (%), missing statistics.

Response 8: Thanks for your carefulness. We measured the rates of diarrhea and soft stools without statistics during the trial, but the total stool score (TFS) can further confirm the accuracy of the diarrhea and soft stool rates.

Comment 9: L515-516. These two sentences should be added to the abstract (end of the abstract).

Response 9: Thank you for your suggestion, we have added these two sentences to the abstract in blue font. Please see line 33-35.

We believe that our responses have well addressed all concerns from you. We hope our revised manuscript can be accepted for publication. And if you have any question about this paper, please don’t hesitate to let us know.

Sincerely yours,

Jinping Deng